# Anticancer Treatment Goals and Prognostic Misperceptions among Advanced Cancer Outpatients

**DOI:** 10.3390/ijerph19106272

**Published:** 2022-05-21

**Authors:** Carlos Eduardo Paiva, Ana Clara Teixeira, Bruna Minto Lourenço, Daniel D’Almeida Preto, Talita Caroline de Oliveira Valentino, Mirella Mingardi, Bianca Sakamoto Ribeiro Paiva

**Affiliations:** 1Palliative Care and Quality of Life Research Group, Post-Graduate Program, Barretos Cancer Hospital, Barretos 14784-400, SP, Brazil; ana.clara.teixeira@hotmail.com (A.C.T.); brunaminto65@gmail.com (B.M.L.); ddpreto@gmail.com (D.D.P.); talitavalentino@gmail.com (T.C.d.O.V.); mirella.mingardi@hcancerbarretos.com.br (M.M.); bsrpaiva@gmail.com (B.S.R.P.); 2Department of Clinical Oncology, Barretos Cancer Hospital, Barretos 14784-400, SP, Brazil; 3Researcher Support Centre, Learning and Research Institute, Barretos Cancer Hospital, Barretos 14784-400, SP, Brazil

**Keywords:** cancer, prognostic markers, palliative care, perception of curability, communication

## Abstract

(1) Background: In the context of cancer incurability, the communication processes involving clinicians and patients with cancer are frequently complex. (2) Methods: A cross-sectional study that investigated outpatients with advanced cancers and their oncologists. Both were interviewed immediately after a medical appointment in which there was disease progression and/or clinical deterioration, and were asked about the patient’s chance of curability and the goals of the prescribed cancer treatment. The patients were asked whether they would like to receive information about prognosis and how they would like to receive it. The analyses of agreement on perceptions were performed using the Kappa’s test. (3) Results: the sample consisted of 90 patients and 28 oncologists. Seventy-eight (87.6%) patients answered that they wanted their oncologist to inform them about their prognosis; only 35.2% (*n* = 31) of them said they received such information at their present appointment. Regarding how they would prefer prognostic disclosure, 61.8% (*n* = 55) mentioned that the oncologist should consider ways to keep the patient’s hope up; 73% (*n* = 65) of the patients reported odds >50% of cure. The agreement between oncologists’ and their patients’ perceptions regarding the treatment goals and curability was slight (k = 0.024 and k = 0.017, respectively). (4) Conclusions: The perceptions of patients and their oncologists regarding the goals of treatment and their chances of cure were in disagreement. New approaches are needed to improve the communication process between oncologists and patients with advanced cancer.

## 1. Introduction

In the context of cancer incurability and palliative care (PC), the communication process involving clinicians and patients with cancer are frequently complex. Discussions about prognosis, preferences, and priorities of care are challenging for the oncologists, so that discussions about topics that may refer to death and dying are postponed until such dialogue is unavoidable [1,2]. Previous studies have shown that 60–90% of patients with life-threatening diseases report that they have never discussed end-of-life care issues with their physician [3,4,5].

Communication issues can negatively influence the patient’s psychosocial experience, symptom management, and treatment decisions, often becoming an impediment to adequate care [6,7]. Thus, the communication of difficult news and the discussion of prognosis are areas that require attention, because what is communicated (and how it is done), will influence how the patient deals with the information provided, reacts, and adjusts to their new reality.

Identifying patients who misunderstand their situation is of paramount importance. Patients who do not have a sufficiently accurate understanding of their disease will hardly be a partner with his/her oncologist while making treatment decisions, and would not be able to perform the “cost–benefit” analysis that leads to an appropriate treatment decision in a palliative setting, so that the patient runs the risk of accepting, or even demanding, aggressive or toxic forms of treatment from which he/she can benefit little [8], thus expressing a preference for treatments that aim to prolong life (but may cause great suffering) over palliative treatment (emphasizing comfort) [9,10,11].

A previous study [12] compared the attitudes and beliefs of PC physicians from South America, Europe, and North America regarding communication with terminally ill patients. When asked if their patients knew about their terminally ill situations, 18%, 93%, and 26% of the physicians from South America, North America, and Europe reported that they did, respectively. Later, a Brazilian study [13] interviewed clinical oncologists, cancer outpatients, and their family members and found that 92% of the patients reported they would like to know about their terminality situation, a percentage higher than that of the physicians and family members themselves. However, this Brazilian study recruited mostly non-metastatic patients with a higher educational level. Furthermore, the study investigated what the patient would like to know, but not what they actually knew. A multicenter study [14] evaluated patient’s perception of curability in a sample of advanced cancer patients followed by PC teams. The assessment of the chances of cure was dichotomous (yes versus no), with Brazilian patients reporting 32.5% curability. Considering that cultural differences are relevant to patients’ perceptions of curability, as well as the timing of the patient’s experience in the continuum of cancer care, no previous Brazilian study has evaluated outpatients with advanced cancers while undergoing systemic cancer treatment and compared them with their physicians’ opinions.

The main aim of this study was to understand what Brazilian patients with incurable advanced cancers know about their prognosis. Furthermore, patients’ opinions about treatment goals (to cure the disease, to increase life span, or to improve quality of life) and perceptions of chance of cure were confronted with the opinions of their oncology physicians.

## 2. Materials and Methods

### 2.1. Study Design and Place of Study

A cross-sectional study conducted at the Department of Clinical Oncology at the Barretos Cancer Hospital (BCH, Barretos, SP, Brazil). BCH is currently considered one of the main references for cancer treatment in Latin America, with public care for more than 6000 patients/day, involving prevention, diagnosis, treatment and palliative care units, as well as cancer teaching and research.

### 2.2. Eligibility Criteria

Adult patients of both genders diagnosed with incurable primary breast, gynecologic, urologic, or gastrointestinal cancer and consulted on the same day by the clinical oncologist for identification of inoperable disease progression or recurrence were included. These pathologies were chosen because the outpatient clinics are located in close proximity to the hospital, making it easier to collect data from patients. Patients with hematologic malignancies; delirium; manifest desire about not talking about prognosis; and any decompensated clinical or psychiatric condition considered risky for study participation according to the clinical oncologist or the researcher were excluded.

Physicians: Included were clinical oncologists, both tenured and resident, working at the Breast and Gynecology, Urology, and Digestive Tract Outpatient Clinics at the BCH, who had consulted patients eligible for the study. Visiting physicians or short-term interns were excluded.

### 2.3. Data Collection

Potentially eligible patients were selected by convenience in clinical oncology outpatient clinics. After checking the patients’ eligibility criteria, they were personally approached by the study researcher after the medical consultation. If they agreed to participate in the study, an ICF was voluntarily signed. Trained researchers conducted face-to-face interviews using structured questionnaires in separate rooms and without the presence of family caregivers. Sociodemographic and clinical characteristics of the patients were obtained from the medical records. For the analyses of this study, patients were interviewed at only one moment.

The study researchers individually approached clinical oncologists from the Breast and Gynecology, Urology, and Digestive Tract Outpatient Clinics and invited them to participate in the research.

The items for the questionnaires (Appendix A) were based on issues identified in previous studies [15,16,17] and developed by the authors to be used in the present research. The clarity and pertinence of each item were evaluated by a committee of experts. Both were tested for understanding in a pilot sample of five cancer patients (Patient Form) and two oncologists (Physician Form). Of the total 19 and 12 items from the patient and physician questionnaires, only 10 (items 3, 5, 8–15) and 2 (items 4 and 12) were used for the present analysis, respectively.

The research clinical data were stored online in REDCap (Research Electronic Data Capture) [18].

#### 2.3.1. Patient Questionnaire

Patients were asked about their communication preference regarding treatment and prognosis. Those who answered that they did not want information about the prognosis were asked to choose one of the following options: “I see no reason to know (would not change my life at all)”, “I think it can do me harm”, “I don’t believe in the life time estimates made by doctors”, or another response. Those who answered that they would like to know information about their prognosis were asked how much information they would like (“I think it is important to know as much detail as possible about my prognosis” or “I think it is important to have a general notion, but without too much detail”). In addition, patients were asked how they would like oncologists to talk about the prognosis (“be realistic”, “be realistic, but try to keep my hope with positive information” or “only worries about keeping my hope with positive information”). Patients were also asked whether they received information about the prognosis at their current appointment (yes vs. no), their perceptions of chance of cure (ranging from 0 to 100%), and about the main treatment goals explained by their oncologists (categorized as “to cure cancer completely”, “to increase the life time as much as possible”, “to relieve the symptoms and to make the patient not suffer” and “not discussed”).

#### 2.3.2. Physician Questionnaire

Although the questionnaire answered by the oncologists contains several items on the assessment of the patients’ prognosis, only two items were used in the present analysis: perceived chance of cure (0%, <10%, 11–24%, 25–49%, 50–74%, 75–90%, and >90%) and information given to the patient about the main treatment aims (as in the Patient Questionnaire).

### 2.4. Statistical Analysis

Patient preferences regarding the communication process were analyzed descriptively. Patients’ perceptions of chance of cure were further categorized as 0%, <10%, 11–24%, 25–49%, 50–74%, 75–90%, and >90%. The agreements between patients and clinical oncologists regarding the chances of cure and perceptions of main goals of treatment were assessed using Kappa’s test. Kappa coefficient interpretation was ‘poor’ (<0.00), ‘slight’ (0.00–0.20), ‘fair’ (0.21–0.40), ‘moderate’ (0.41–0.60), ‘substantial’ (0.61–0.80), and ‘almost perfect’ (0.81–1.00) [19]. Assuming that the categories were ordered and accounting for the distance between them, the weighted kappa was also calculated. The Statistical Package for the Social Sciences (SPSS; version 20.0, Chicago, IL, USA) was used for data analysis. For the study, the significance level was defined as 0.05.

## 3. Results

### 3.1. Participant’s Characteristics

Between April 2021 and September 2021, 151 patients were potentially eligible for the study; however, 45 were excluded (*n* = 12, inability to answer the questionnaires; *n* = 33, unavailability of study personal to collect data), and 16 refused to participate. Thus, the sample for this study consisted of 90 patients (enrollment rate = 59.6%). Table 1 shows the main characteristics of the study patients.

Twenty-eight oncology physicians were included in the study (enrollment rate = 100%). Of these, 11 were tenured clinical oncologists and 17 were clinical oncology residents. The median (p25–p75) age of the physicians was 34 (29–43) years.

### 3.2. Patient Preferences Regarding Communication of Prognosis

Patients were asked how much information they would like to receive from their oncologist regarding their cancer treatment. In response, 79.8% (*n* = 71) answered they preferred to know all the details and 19.1% (*n* = 17) answered they preferred to know only the main information. Not wanting to know anything about the treatment was the choice of only one patient (*n* = 1, 1.1%).

Seventy-eight (*n* = 78, 87.6%) patients said they wanted their oncologist to inform them about their prognosis (chances of cure and/or remaining life expectancy). On the other hand, 10 patients (11.2%) reported not wanting to receive information about prognosis; when asked about the reasons for this, one patient (1 out of 10) answered “I don’t see any reason to know (it wouldn’t change my life)”, four (4 out of 10) patients answered “I think it could do me harm” and the rest (5 out of 10) did not want to answer or gave another response. Regarding those patients who answered it was important to know their prognosis (*n* = 78), 61.5% (48 out of 78) thought it was important to know as many details as possible, and 35.9% (28 out of 78) thought it was only important to have a general idea without many details.

The patients were asked how they would prefer the oncologist to explain the prognosis to them. Thirty-three (37.1%) participants responded that the oncologist should be realistic; 40 (44.9%) said that the oncologist should be realistic but try to maintain the patient’s hope; and 15 (16.9%) responded that the oncologist should only be concerned with maintaining the patient’s hope with positive information.

Although most patients reported wanting information about prognosis, only 31 patients (35.2%) said they received such information at their present appointment. Of the patients who received information about prognosis (*n* = 31), 93.5% (*n* = 29) of them responded that they received enough information, one claimed for more information, and another patient refused to answer this question.

### 3.3. Agreement between the Patient’s and Oncologist’s Opnions about Tratment Goals and Prognosis

Regarding the goals of treatment, no oncologist explained to patients that it would be to cure the cancer, since the cases included were all considered incurable. However, 15.5% of the patients (*n* = 13) answered that the main goal of treatment would be to cure them. On the other hand, no patient said that the goal of the treatment would be to relieve symptoms and make them pain-free; according to the oncologists, this option was explained to the patients in 40.5% (*n* = 34) of the times. A slight agreement coefficient was evidenced between oncologists’ and their patients’ opinions regarding the treatment goals (kappa = 0.099, SE = 0.052, *p* = 0.041; weighted kappa = 0.154; Table 2).

The agreement between patients’ and oncologists’ perceptions of chances of cure are described in Table 3. The lack of agreement is evident, since most physicians’ perceptions are of 0% chance of cure (*n* = 77, 86.5%), with few cases with chances <50% (*n* = 9, 10.1%). Regarding patients, 65 (73.0%) reported odds >50% of cure (kappa = 0.017, SE = 0.008, *p*-value = 0.341; weighted kappa = 0.017; Table 3). The median (p25–p75) of patients’ perceptions of chance of curability was 69.5% (35.5–100%).

## 4. Discussion

In this study, in a sample of patients with solid incurable tumors, we found that almost 90% of them would like to receive information about their prognosis; however, a little more than a third of them reported having received such information in the medical appointment. The patients’ perceptions about the goals of treatment and their chance of cure were very different from those reported by the oncologists who consulted them. Attention must be given to improving the communication process in patients with advanced cancers undergoing oncologic treatment.

Approximately 70–90% of advanced cancer patients wish to discuss their prognosis with their healthcare providers [1,15]. In the Coping with Cancer Study [1], 71% of the patients wanted to know their life expectancy but only 18% recalled a prognostic disclosure. Our results were similar, since 87.6% wanted to know their prognosis but only 35.2% reported having received such information. Among the many reasons why patients desire prognostic disclosure is the possibility to relieve the patient and family of the burden of unnecessary treatments [20]. This reason seems quite relevant to us in an outpatient oncology setting, where patients with prognostic misperceptions may demand to receive potentially futile treatments.

Prognostication is a dynamic and multi-step process. Along the prognostic continuum are prognostic prediction, prognostic disclosure, prognostic awareness, prognostic acceptance, and prognostic-based decision making [21]. In other words, after having communicated about prognosis, patients need to understand the predictions presented (prognostic awareness) and then process these expectations emotionally (prognostic acceptance). Many personal and cultural aspects are probably involved in regulating this process. Patients with advanced cancer commonly hesitate to ask directly about their prognosis and instead wait for their oncologists to start such a conversation [22]. On the other hand, oncologists are often reluctant to openly discuss the prognosis for the fear of reducing patients’ hope and causing psychological problems [1]. To make this clinical situation even more relevant, patients’ prognostic misperceptions have been associated with aggressive end-of-life care and inadequate treatment decisions [11,23].

A systematic review found that only 49% of patients with advanced cancer had an accurate prognostic awareness [24]. In fact, more than two thirds of patients with incurable solid tumors thought their palliative chemotherapy [25], radiation [26], and/or surgery [27] could cure them. In a previous multicenter study [14], Brazilian patients with advanced cancer receiving PC, recruited also in the BCH, reported 32.5% chance of cure, which indicates a more realistic notion of their clinical situations when compared to our results. However, unlike the previous study, the patients from our study are at an earlier stage of cancer evolution, still receiving systemic oncologic treatment, which may modify the patients’ prognostic interpretation. To our knowledge, no other Brazilian study to date has evaluated the perception of curability of patients with advanced cancer. In the real world, the clinical oncologist often has to decide whether to start, stop, or switch antineoplastic treatment while evaluating patients with advanced cancer. When the patient is in good clinical condition and desires further treatment, the therapeutic decision is made easily. However, when the patient is in borderline clinical conditions, due to poor performance status or significant comorbidities, for example, patient participation in the informed decision-making is of utmost importance. Prognostic misperceptions or the lack of acceptance of the prognosis will not allow decision making as it should be.

In general, oncology consultations are rapid and focused on the response to oncology treatment. According to the TEAM (time, education, assessment, and management) approach [28], an additional monthly time of 1 hour/per patient is needed to include other approaches to patient care, particularly education about prognostic awareness and discussions about realistic treatment options. Every time of treatment response assessment, particularly when the disease is progressing, is a time to review goals of care. The American Society of Clinical Oncology (ASCO) recommends that patients with advanced cancer should be referred to palliative care within 8 weeks of diagnosis of an advanced disease [29]. However, we know that patient demand is certainly greater than the availability of specialized palliative care services in many regions (Latin America, for instance). Thus, in parallel with the increase in the number of palliative care specialists and teams, we must foster palliative care education at the primary and secondary levels. In addition, each cancer hospital must seek its ideal structure for integration with palliative care. One way to accomplish this integration is through the training of clinical oncologists in how to conduct difficult news communications and to perform goals of care consultations. In fact, when patients are aware of their disease and prognosis and, preferably, emotionally accept the situation, we believe that the decisions to treat or not to treat are facilitated with the possibility of sharing decisions in a mature way and with less patient/family distress.

It is important to emphasize that cancer patients, even being aware of the presence of distant metastases, may have exaggeratedly optimistic expectations about the efficacy of antineoplastic treatment as a coping strategy, avoiding thinking about mortality [30,31,32]. Interestingly, in our study, when asked how they would like to be informed about the prognosis, 61.8% of patients reported that the oncologist should consider a way to maintain hope during communication. In this regard, 16.9% of the patients reported that keeping hope was even more important than telling the truth. In a previous qualitative study [33], ongoing chemotherapy and receiving good news from oncologists were ways to encourage hope. Patients appreciated the oncologist to be honest, but within positive limits. It is possible that the denial of incurability is fed by hope [34]. In fact, hope is the main element of adaptive strategies and should be taken in consideration by oncologists while conducting prognostic disclosure [35].

A previous study [36] classified patients with advanced cancer into three groups: those who did not receive palliative radiation or palliative chemotherapy (“no-treatment group”), those who received it and correctly believed the intent was palliative (“accurate group”), and those who received it and mistakenly believed the intent was curative (“misperception group”). Prognostic understanding scores were higher in the no-treatment group compared to the misperception group, but lower than the accurate group. Thus, a portion of patients with inaccurate perception of treatment goals in fact have inadequate understanding of their prognosis as a whole. In our study, no patient reported that the goal was to alleviate symptoms; on the other hand, this was the most frequent response among oncologists. There was no agreement between the answers of patients and their oncologists regarding the treatment goals, which raises concerns about the communication process in this patient profile, when performed by the clinical oncologist in daily practice.

The study has several limitations. One limitation is the way perceived curability was measured on a scale of 0–100%. Several factors are involved in prognostic awareness, and it is probably a multidimensional concept, involving cognitive understanding of prognosis, emotional coping, and adaptive response [37]. Subsequent studies should use other tools to measure perceived curability, necessarily involving cognitive awareness itself and acceptance of prognosis [21]. Although the assessment items used in the study have not been formally validated, they have undergone content evaluation and pilot testing before the study began. Another limitation was that we did not investigate the comorbidities of the patients. It is possible that severe comorbidities may have interfered with the patients’ curability perception. In addition, social support was not measured in the present study. In this sense, social support may be influencing the interpretation and acceptance of the prognosis, as well as the prognosis itself. Another possible limitation is that the study population included more women than men, due to the higher proportion of breast cancer patients and the non-inclusion of patients with lung and head and neck cancers. Thus, the results of the study need to be interpreted considering possible sampling bias. However, the authors consider that the misperception of prognosis is something of the local culture and education of patients and physicians and less related to the type of tumor specifically.

This study has some practical implications. The first one is the identification of the need for training of oncology staff in more effective forms of communication. In a recently published systematic review [38], the authors identified nine different interventions with potential impact on prognostic understanding. However, even after the use of such interventions, a significant portion of the patients maintained inaccurate perceptions of their prognosis. In addition, it is important to emphasize that cultural issues specific to Brazil, as well as the moment in which patients are located in the advanced cancer care continuum, are characteristics that may be influencing the findings of the study. In this sense, further studies are needed to identify best education strategies able to improve patients’ prognostic awareness and acceptance.

## 5. Conclusions

Almost 90% of patients would like to receive information about their prognosis, yet just over a third of them end up receiving such information at their medical appointment. There is no agreement between the perceptions of patients and their oncologists regarding the goals of treatment and their chances of cure. New approaches are needed to improve the communication process between oncologists and patients with advanced cancer.

## Figures and Tables

**Table 1 ijerph-19-06272-t001:** Characteristics of the patients included in the study.

Characteristics	*n*	%
Age, median (min–max)	59	32–83
Gender		
Male	19	21.1
Female	71	78.9
Years of education		
<8	36	40.0
8–11	10	11.1
>11	44	48.9
Marital status		
Single	12	13.8
Married	57	65.5
Divorced	8	9.2
Widower	10	11.5
Missing	3	-
Family income (minimum wages)		
1 or less	30	33.3
2 to 3	40	44.4
More than 4	20	22.2
Concurrent Palliative Care		
Yes	14	15.6
No	73	81.1
Missing	3	3.3
Primary cancer type		
Urological	12	13.3
Gynecological	13	14.4
Breast	43	47.8
Digestive (non-colorectal)	8	8.9
Colorectal	14	15.6
Treatment indicated in the current appointment		
Chemotherapy	51	56.7
Hormone therapy	16	17.8
Palliative Care only	8	8.9
Other	15	16.7
Palliative Treatment line		
1st line	26	28.9
2nd line	20	22.2
3rd line or more	27	30.0
Not applicable	15	16.7
Missing	2	2.2
ECOG-PS		
0	20	22.2
1	43	47.8
2	17	18.9
3	8	8.9
4	2	2.2

**Table 2 ijerph-19-06272-t002:** Analysis of patient–oncologist agreement on primary treatment goals.

	Oncologists	To Cure Cancer Completely	To Increase the Life Time as Much as Possible	To Relieve the Symptoms	Other Answer	Total
Patients						
To cure cancer completely		0	7	5	1	13
To increase the life time as much as possible		0	29	19	3	51
To relieve the symptoms		0	0	0	0	0
Other answer		0	6	10	4	20
Total		0	42	34	8	84

The cells describing the agreement between the opinions are highlighted in gray.

**Table 3 ijerph-19-06272-t003:** Agreement between patients’ and oncologists’ perceptions of chance of cure.

Chance of Cure (%)		Oncologists						
		0	<10	11–24	25–49	50–74	75–90	>90
Patients	0	13	0	0	0	0	0	0
	<10	1	0	0	0	0	0	0
	11–24	3	0	0	0	0	0	0
	25–49	7	0	0	0	0	0	0
	50–74	19	2	0	0	0	0	0
	75–90	12	0	0	0	0	0	0
	>90	22	6	2	1	1	0	0

The cells describing the agreement between the opinions are highlighted in gray.

## Data Availability

The data presented in this study are available on request from the corresponding author. The data are not publicly available due to ethical reasons.

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
