# Peer review of "Anticancer Treatment Goals and Prognostic Misperceptions among Advanced Cancer Outpatients"

_ijerph, 2022, doi:10.3390/ijerph19106272_

Round 1

Reviewer 1 Report

Article with interest for the ambit of communication in health especially in the ambit of the communication of the bad news. I put here some points for improvement:

1) the authors do not make reference to the approval of the research project by an ethics committee. Is this an oversight? 
2) The authors mention in lines 82 to 88 that they interviewed the patients. I don't understand what methodology they used to analyse their content. What type of interview was carried out? Structured? Semi-structured? Open-ended? For the sake of the transperency of the scientific work, you should refer the methodology of the analysis of the interview content and the type of interview used.
3) In the conclusions section, the authors are very brief. Taking into account the data obtained in this research, despite the limitations, what consequences can they refer to for the training of health professionals, what consequences do these data have for the practice of health care and what research should be continued to increase the evidence?

Reviewer 2 Report

The study is a a cross-sectional study that investigated 16 outpatients with advanced cancers and their oncologists. Both were interviewed immediately after a medical 17 appointment in which there has been disease progression and/or clinical deterioration, and were asked about 18 the patient's chance of curability and the goals of the prescribed cancer treatment.

However there several problems with the design of the study:

  1. In the introduction it is not clear why was this study undertaken? Have similar studies been done in Brazil before? Why is this study special in comparison to other studies? This paper is presenting only patients data but is clear that the research took place among patients and oncologists why was the oncologist data included from the study (at least that is what is stated in the abstract) ?
  2. In the methodology section it is not clear what is the status of the hospital where the research took place is it regional  big or small hospital how many cancer patients do they have per year what is the actual patient and physician population in the hospital( how many cancer patients, how many oncologists). What is actual response rate? How many patients were at the time of the study in the hospital and how many decided to participate? The same for the oncologists?
  3. Why did the researchers decided to do quantitative instead of qualitative study? What type of questionnaire did they use? How was the questionnaire constructed and validated? Was it taken from another study?  What statistical methods were used (in detail) to analyse the data?
  4. The questionnaire should be described in detail?
  5. 5. In discussion what is the difference and similarities of your study with similar studies in Brazil and in the world?

Reviewer 3 Report

Congratulations on the job. However, it presents limitations at the methodological level that are already indicated in its limitations. They should value the objective of the work and value the use of specific tools. I recommend an assessment of the state of the question to see the situation in other studies.

The study presents little new information. Regarding the introduction, the presence of previous similar studies should be assessed. The objective is not clear and is too broad. Regarding material and methods, the inclusion criteria of the patients are not justified. Is there family support? Are there different pathology knowledge conditions? Regarding the variables, what questions have been asked exactly? Is there a validated scale? Has a data collection form been used? In results, lines 125-132 literally repeat the information in table 1. The wording of the results is not clear. The limitations of the study already indicate that perhaps another approach should have been used.

Round 2

Reviewer 2 Report

The paper has been improved according to suggestions by the reviewer. It should be published. 

Reviewer 3 Report

The article has improved although methodological problems persist that I believe start in the initial development of the work.
I still do not understand that the selection of pathologies is due to proximity to the hospital when it could be a factor that affects quality of life. Sample selection for convenience can be a limiting factor. There is also no mention of family support or comorbidity.
It is not understood why you have not used all the articles (lines 116-118). Has the level of health literacy been assessed? Is the sample homogeneous?
Regarding the questionnaire answered by the oncologists, what criteria have been used to select the items? Why have they been included in this study?
Why have validated tools not been used for a quantitative study?
The categorization of responses is not homogeneous.
